# Monitoring Strain Response of Epoxy Resin during Curing and Cooling Using an Embedded Strain Gauge

**DOI:** 10.3390/s21010172

**Published:** 2020-12-29

**Authors:** Hongyu Dong, Huiming Liu, Arata Nishimura, Zhixiong Wu, Hengcheng Zhang, Yemao Han, Tao Wang, Yongguang Wang, Chuanjun Huang, Laifeng Li

**Affiliations:** 1State Key Laboratory of Technologies in Space Cryogenic Propellants, Key Laboratory of Cryogenics, Technical Institute of Physics and Chemistry, Chinese Academy of Sciences, Beijing 100190, China; donghongyu17@mails.ucas.ac.cn (H.D.); huimingliu@mail.ipc.ac.cn (H.L.); nishimura.arata@toki-fs.jp (A.N.); zxwu@mail.ipc.ac.cn (Z.W.); zhanghengcheng@mail.ipc.ac.cn (H.Z.); hanym@mail.ipc.ac.cn (Y.H.); wangtao@mail.ipc.ac.cn (T.W.); wangyongguang16@mails.ucas.ac.cn (Y.W.); 2School of Future Technology, University of Chinese Academy of Sciences, Beijing 100049, China

**Keywords:** epoxy resin, strain response, curing process, thermal shrinkage

## Abstract

The present work describes the monitoring system of the real-time strain response on the curing process of epoxy resin from the initial point of curing to the end, and the change in strain during temperature changes. A simple mould was designed to embed the strain gauge, thermometer, and quartz standard sample into the epoxy resin, so that the strain and the temperature were simultaneously measured and recorded. A cryogenic-grade epoxy resin was tested and the Differential Scanning Calorimetry (DSC) was used to analyse the curing process. Based on the DSC results, three curing processes were adopted to investigate their influence on strain response as well as residual strain of the epoxy resin. Moreover, impact strength of the epoxy resin with various curing temperatures were tested and the results indicate that the curing plays a crucial role on the mechanical properties. The method will find cryogenic application of epoxy adhesives and epoxy resin based composites to monitor the strain during the curing process as well as the cryogenic service.

## 1. Introduction

The epoxy resins have been widely used as an adhesive, a matrix material of fiber-reinforced plastics and an impregnating material for large-scale superconducting magnets because of their good mechanical, electrical insulating, and easy-to-fabrication properties. In general, the adhesives, the fiber-reinforced plastics and the impregnating material will be used at cryogenic temperatures. The thermal and the mechanical performance of epoxy resins will control the mechanical behaviour of the adhesive, the fiber-reinforced plastics, and the reliability and stability of the superconducting magnets. The volumetric change leads to potential inconvenience such as dimensional inaccuracy [1], residual stress development, and surface finish imperfection in reinforced epoxy resins, and then leads to degradation of the performance. The shrinkage of the resin determines the surface integrity of the composite structure. Residual stresses can lead to warping, loss of mechanical properties, and premature debonding in adhesive joints [2], laminated composites structures [3], and large 3-D printed parts [4]. Therefore, it is necessary to measure and track the resin shrinkage during the curing process in order to minimize the surface failure and produce the good surface quality [5]. The volumetric change of thermoset resin results from two factors, i.e., curing shrinkage caused by the chemical cure reaction, and the thermal shrinkage during cooling down to cryogenic temperature [6,7]. Chemical reaction affects the volumetric cure shrinkage of thermoset resin which occurs during the chemical network formation. It happens as the result of the change from Vander Waals links to covalent links among the molecules [7,8,9].

Since the residual stresses generated during both curing and temperature change lead to the reduction of performance and the premature failure of components, it is of great significance to carry out the quantitative measurement of the volumetric change during curing process and cooling down to cryogenic temperature. Curing shrinkage occurs while the resins change from the liquid phase to the rubbery state and finally to glass state, which makes the shrinkage measurement difficult during the whole curing process.

The curing shrinkage can be measured with several methods, which mainly fall into two categories, i.e., the direct methods and the indirect methods. For example, water or mercury dilatometers are the most common instruments used to measure the shrinkage based on the volume dilatometry. Pycnometers are another simple method to measure the volumetric change in a dry state, and it allows only the glass state rather than the development during the curing process. Video-imaging is another method to analyse the volumetric shrinkage of the epoxy resin during the curing process. Non-volumetric dilatometry methods, which can be categorized to the indirect methods, usually measure the one-dimensional shrinkage and the volumetric shrinkage under various assumptions. Several non-volumetric methods have been developed, such as the shadow Moire method, the dynamic mechanical analysis (DMA), online monitoring using linear variable differential transformer (LVDT) transducers and optical sensors embedded in the epoxy resin, the modified rheology method, and the gravimetric method.

The thermal expansion behavior in the cryogenic temperature range is one of the most important thermophysical parameters for the design of and the development of thermal components. This is because the mismatch of coefficients of thermal expansion (CTEs) α between the epoxy resin and the combined materials will inevitably create the thermal stress on the interfaces due to the temperature variation or the temperature gradient. Calculation of the thermal stress with the temperature variation or the temperature gradient requires the accurate CTE or the thermal expansion data. The quantitative value of the thermal stress relates to several materials properties including the CTE, Young’s modulus, Poisson’s ratio, and the value of the temperature variation or the temperature gradient. To measure the thermal expansion or the CTE at cryogenic temperatures, various dilatometer approaches (such as capacitive dilatometers, interferometric dilatometers, and inductive dilatometers), transformer techniques, strain gauge techniques, and optical speckle photography techniques have been developed during the past decades [10,11,12,13]. All developed methods are generally divided into two categories, i.e., the absolute deformation and the relative deformation (strain) methods. On the other hand, the resolution of the deformation measurement has been significantly improved. Even the modified dilatometers with high-resolution of 2 × 10^−5^ nm was developed with a SQUID sensing element [14]. To investigate the thermal expansion or the CTE at cryogenic temperatures, most of these techniques require the complicated strain or the displacement sensing elements, as well as the complicated and controllable cryogenic system. This is because the CTE of most materials become much smaller at cryogenic temperature and thus needs more sensitive methods. At room temperature, the average CTE of typical metallic and polymeric materials is in magnitude of 10^−5^ K^−1^, thus the measurement methods with a sensitivity of Δl/l~10^−6^ and with the temperature interval of 10 K will give an inaccuracy of 1% or less. This sensitivity can be achieved easily by traditional dilatometers [15]. However, the CTE of copper becomes small at cryogenic temperatures, for example, 10^−7^ K^−1^ and 5 × 10^−9^ K^−1^ at 15 K and 5 K, respectively, and it requires sensing elements with high sensitivity. On the other hand, a complicated special cryostat, which is generally cooled by cryogenic liquids like liquid helium (4.2 K) or commercially available refrigerators, requires the CTE down to low temperature than 77 K [16]. Moreover, the cryostat is designed and assembled only by the researchers and there is no commercial apparatus. Therefore, a few studies investigated the thermal expansion behavior with the liquid nitrogen but not liquid hydrogen or lower.

Strain responses of curing stress have been studied by scientists by using carbon fibers [17,18,19,20], glass fibers [21], and carbon nanotube yarn [22]. For example, Y. Huang et al. used Raman spectroscopy to study the residual stress in carbon fiber/epoxy resin composites and made it clear that this technology is an excellent scientific tool for determining the residual stress in composites [23]. Omar Rodríguez-Uicab et al. have investigated the curing effects and development of residual stresses during epoxy resin curing through the electrical response of a single carbon nanotube yarn embedded in the epoxy polymer [22]. These landmark works boost the development of multifunctional smart materials integrated into structural composites, and have made extraordinary contributions to the study of resin curing kinetics.

The embedded strain gauge technique may require considerable effort in the initial lay-out of the embedding frame but it utilises the simplicity and the accuracy of strain gauge measurement method for static or dynamic conditions [24]. M.T. Hillery et al. have embedded strain gauges at some points along the length of a lead rod, pulling the rod through a suitably modelled epoxy die, and obtained a number of strain values in the axial, radial and hoop direction [25]. E. Schnack et al. have applied embedded strain gauges in the fiber/epoxy laminates to determine interlaminar stresses in carbon fiber/epoxy composites [26]. M. Kanerva et al. have embedded electrical resistance strain gauges in the hybrid material which were laminated using carbon-fiber-reinforced plastic (CFRP) and tungsten to measure the thermal expansion (residual strain) of the laminates and concluded that the embedded electrical resistance strain gauges can be used to determine the thermal expansion of a hybrid laminate [27].

Although both the curing shrinkage during the curing process and the thermal shrinkage during the cooling down to cryogenic temperatures have been extensively, separately measured, the measurement of them with an identical method has rarely been achieved. In the present work, a special structure was designed to embed the electrical strain gauges into the epoxy resin, with which the curing shrinkage and the thermal shrinkage of a cryogenic-grade epoxy resin were investigated. The cryogenic thermal expansion coefficient measured by embedded strain gauge were compared with the measurements using surface-bonded strain gauges. The relationship between the impact performance of the epoxy resin and the residual strain of curing was explored.

## 2. Materials and Methods

### 2.1. Cryogenic-Grade Epoxy Resin

The cryogenic-grade epoxy resin system consists of di-glycidyl ether bisphenol F (DGEBF) and diethyltoluenediamine (DETDA) as curing agent. The weight ratio of the DGEBF and DETDA is 100:24.

### 2.2. Cryogenic-Grade Strain Gauge

Strain gauge (Tokyo model CFLA-1-350-11, with a resistance of 350 Ω) was used in the work. The grating pitch of metal sensitive grid of the strain gauge is 1 mm, and the wide applicable temperature of the strain gauge is reported to be 4 K to 353 K. The principle of this work was Wheatstone bridge (shown in Figure 1), and the two-gauge method was used.

When the bridge is balanced, the electric potentials of point B and point C are equal. R_x_ can be calculated by:(1)$$R_{x}=\frac{R_{1}}{R_{3}}\times R_{2}$$
where the resistance value of $R_{x}$ represents the strain gauge embedded in the epoxy; $R_{2}$ represents the strain output of quartz with a strain gauge bonded on the surface; $R_{1}$ and $R_{3}$ are strain gauges with resistance of 350 Ω.

### 2.3. Experimental Method

Differential Scanning Calorimetry (DSC, NETZSCH model DSC 404 F3, Erich NETZSCH GmbH & Co. Holding KG, Selb, Germany) was used in the present work in the 298–523 K temperature range, and the rate was 283 K/min. Samples are taken out from oven every hour and were tested with DSC to track the cured percentage at each stage. From the DSC curve of the resin curing reaction, the size of the exothermic peak and the heat of the curing reaction can be obtained. Based on the DSC result, three curing processes, i.e., 353 K/24 h + 403 K/12 h (labelled as Pro. I hereafter), 353 K/12 h + 403 K/12 h (labelled as Pro. II hereafter), and 343 K/24 h + 423 K/12 h (labelled as Pro. III hereafter) were used in this work and then the influence of the curing process and the curing shrinkage was investigated. The degree of reaction α can be calculated with exotherm by Formula (2):(2)$$\alpha =\frac{\Delta H_{0}-\Delta H_{R}}{\Delta H_{0}}\times 100\%$$
where $\Delta H_{0}$ is the total heat released when the resin is completely cured, and $\Delta H_{R}$ is residual heat of reaction after resin curing.

The strain response during the curing process was measured using a strain gauge meter, a temperature monitor, and an oven with the maximum heating temperature of 573 K, as shown in Figure 2a. The system enables simultaneous recording of the real-time strain response and the temperature of the cryogenic-grade epoxy resin during the curing process. According to Standard ISO 8130–6:1992 [28], the initial solid point in this work was determined by manual methods based on evaluation of the rheological behavior of the epoxy resin during curing, the pattern of the resin fiber drawing to be probed [29]. The timer is started after putting the fully mixed components into the oven. As time goes by, the viscosity of the mixed glue becomes larger, and the wire drawing gradually begins [30]. The time when the glue fails to draw wire is taken and it is recorded as the initial solid point of the epoxy resin [31]. Moreover, the residual strain during cooling down of the epoxy resin from the elevated temperature to room temperature can be obtained. To investigate the influence of boundary condition on the curing behavior, two kinds of moulds were used in the experiments, i.e., the rigid boundary and the soft boundary. The rigid boundary is obtained with a stainless steel mould with a diameter and length of 60 mm and 40 mm, respectively, as shown in Figure 2b. Whereas the soft boundary is obtained with a silicone mould with the same geometry, as shown in Figure 2c. The lids of moulds, whose diameter is 100 mm, are made of copper. There are three parts attached to the lid, including one copper container for a reference quartz (diameter of 7 mm and length of 37 mm), one copper container for Pt-100 thermometer (diameter of 4 mm and length of 37 mm), and one free strain gauge, which was inserted vertically into the epoxy resin. The same strain gauge was bonded to the quartz for reference. The Pt-100 in the copper container was used to measure the temperature inside the epoxy resin. The free strain gauge was used to monitor the real-time strain response of the epoxy resin. Moreover, another Pt-100 thermometer was installed into the chamber containing the mould to monitor the environmental temperature.

To investigate the thermal shrinkage during cooling down to cryogenic temperature of the epoxy resin with the strain gauge, a device cooled by cryocoolers was designed, as shown in Figure 3. The system consists of one cryogenic chamber, one vacuum chamber, and two GM cryocoolers. The sample can be cooled down to around 15 K. The thermal shrinkage of the epoxy resin was measured by a physical property measurement system (PPMS, Quantum Design model PPMS-14, Quantum Design, Inc. (QD), San Diego, CA, USA) with a surface bonded strain gauge on the epoxy resin, whereas the thermal shrinkage was measured with an embedded strain gauge developed in this study.

To investigate the influence of the curing process and the curing residual strain on the mechanical properties of the epoxy resin, the Charpy impact test was conducted at room and liquid nitrogen temperatures. According to ASTM A370 [32], the geometry of specimens for Charpy impact testing is 10 mm × 10 mm × 55 mm. The sketch of notch on the specimen is shown in Figure 4. The notch depth α is 0.2 mm, the notch root radius ρ is 0.05 mm, and the notch angle θ is 45 degree. Five specimens with a geometry of 10 mm × 10 mm × 55 mm for each condition were tested. The average and the standard deviation were calculated. Moreover, the morphology of the fracture surface tested at room and liquid nitrogen temperatures were investigated with a scanning electron microscope (SEM, Hitachi Model S-4800, Hitachi, Ltd., Marunouchi, Chiyoda-ku, Tokyo, Japan). Prior to the observation, gold vaper deposition was carried out on the fracture surface.

## 3. Results and Discussion

### 3.1. DSC Curve and Strain Response of Epoxy Resin during Curing Process

The results of DSC measurement of the epoxy resin cured with different processes are shown in Figure 5a–c. The degrees of curing for different processes are shown in Figure 6a–c and the different final cured percentages are displayed in Table 1. Thus it can be considered that the epoxy resin with different curing process will result in different cured percentage. Moreover, it is observed that the initial solid points of the epoxy resin are around 80% of the resin cured degree for all curing processes.

The real-time strain responses of epoxy in the three curing processes are shown in Figure 6a–d. All the results are obtained from five experiments. Apparent data is the actual output of the real--time strain response of the strain gauge. Both the expansion and shrinkage of the metal wire of the strain gauge and the strain change of the epoxy simultaneously constitute the apparent data. Quartz has been applied as the temperature compensation sample in this experiment. The strain response data after the subtraction of the strain data of the compensation (i.e., temperature compensation) is the compensated data shown in Figure 6a–c.

As shown in Figure 6a–c, points a/a’/a” to point b/b’/b” is the first stage of the curing process, where temperature rises up from room temperature (around 300 K) to the first curing temperature (343 K, or 353 K). During this stage, the epoxy system is very soft with low viscosity. The strain gauge expands freely. The slope shows the thermal expansion coefficient of the strain gauge. From points b/b’/b” to points c/c’/c”, the strain maintains during this period as the temperature is unchanged. Micro curing occurs during this period without global changes. Before points c/c’/c”, curing already proceeds, and cured percentage increases. Micro cured molecules are floating in the soft resin which is not cured yet. At points c/c’/c”, the global change is observed. The micro cured molecules gather resulting in producing bigger molecules and this is the initial point of the epoxy resin to become solid. From points c/c’/c” to point d/d’/d”, the global curing proceeds during this period. The volume decreases, and the strain decreases. From points d/d’/d” to points f/f’/f”, the cured ratio line becomes almost flat which means the curing is coming to end. Thermal expansion of the epoxy and the gauge occur due to the temperature rise. Then the curing proceeds under the constant temperature and the shrinkage of the epoxy occurs. Thus the strain decreases. The last procedure is from point f/f’/f” to point g/g’/g”. Temperature goes down during this period. Both the cooling shrinkage of the metal wire of the strain gauge and the epoxy cooling shrinkage simultaneously cause the strain decreases.

Based on results shown in Figure 6d, the curing residual strains of the epoxy resin with different curing processes can be obtained and shown in Table 1. These results confirm that different curing processes result in different curing residual strain.

### 3.2. Strain Response of the Epoxy Resin Curing with Different Boundary Conditions

To investigate the influence of the boundary conditions on the strain response, two different moulds were prepared, i.e., a stainless steel mould and a silicon mould. The stainless steel mould has a rigid boundary condition and the free shrinkage is allowable only in one direction, whereas the silicon mould has a soft boundary condition which can allow the free shrinkage in all directions. The strain responses of the epoxy resin with different curing processes in the stainless steel mould and the silicon mould are shown in Figure 7a–c. It is noticed that the real time strain response in the two moulds is not identical at each point of time. In case of silicon container, the change in strain output is smaller than that in stainless steel and the reason is the free shrinkage. However, the boundary condition does not affect the final residual strain.

### 3.3. Strain Response of Epoxy Resin during Cooling down Process

Thermal shrinkages of epoxy resin with embedded and surface bonded strain gauges during cooling down process are shown in Figure 8. The results indicate that the real--time strain response monitored by the embedding method mentioned during the cooling down to cryogenic temperature is effective and reliable. While slight differences can be observed from the embedded and bonded lines. The reason is that the surface gauge will be cooled or warmed faster than the inside (embedded). Naturally, the inside will be cooled or warmed slower than the surface.

### 3.4. Mechanical Property and Fracture Morphologies of Epoxy Resin Curing with Different Process

To investigate the influence of curing process as well as residual strain on mechanical property of epoxy resin, the Charpy impact test was conducted at both room and liquid nitrogen temperatures. The impact strengths of epoxy resins with three curing processes is shown in Table 2. These results indicate that epoxy resin with low residual strain resulting from curing process result in higher impact strength than that with high residual strain at both room and cryogenic temperature. In addition, the impact strength of the epoxy resin at cryogenic temperature is lower than that at room temperature, which is consistent with common studies.

The SEM images of epoxy resin cured with different processes and impact fractured at both room and cryogenic temperatures are shown in Figure 9a–f. It can be seen that surface with many facets and cracks for the epoxy resin with low residual strain resulting from curing process, as shown in Figure 8. When comparing surface morphologies of the epoxy resin fractured at room temperature with those fractured at cryogenic temperature, it is observed that the former demonstrate more facets and cracks than the latter. This is consistent with that of Charpy impact strength.

### 3.5. Discussion

Compared with previous research, this work focus on the morphological changes and mechanical properties analysis of the epoxy resin during the curing process. G. A. George et al. studied the cure of TGDDM/DDS epoxy resin by using a microcapillary cell and silica fiber optics coupled to a remote FT--IR spectrometer operating in the near--IR. From a comparison of the changes in the functional groups with time of cure, i.e., loss of epoxide, loss of primary amine, and growth of hydroxyl, real--time plots of the extent of cure of resin cure up to the gel point can be obtained [33]. DSC method was applied in this work to monitor the curing percentage of epoxy during the entire curing process, so it enabled a more complete curing degree curve. Sumit Gupta et al. proposed a noncontact, noninvasive, imaging technique for monitoring epoxy curing. The change in dielectric property (i.e., electrical permittivity) of epoxy specimens subjected to different curing times was captured by electrical capacitance tomography [34]. While in this work, the embedded strain gauge inside the epoxy resin can sense more real changes in the epoxy resin state, including strain changes and temperature changes.The strain gauge can keep track of the internal properties of the epoxy resin after the curing process complete.

## 4. Conclusions

In the present work, embedded foil strain gauges have been successfully applied for monitoring strain response of epoxy resin during both curing process from initial solid point and cooling down to cryogenic process. The strain response test platform using embedded strain gauges was first designed for monitoring the curing process of epoxy resin. Moreover, the reliability of the real--time strain response of the embedded strain gauge method is verified by comparison with results obtained with the surface bonded strain gauge method. Based on the monitoring of strain response during curing process, the residual strain was obtained for different curing processes and its influence on mechanical property was investigated through the Charpy impact test and then verified with photographic analysis.

## Figures and Tables

**Figure 1 sensors-21-00172-f001:**
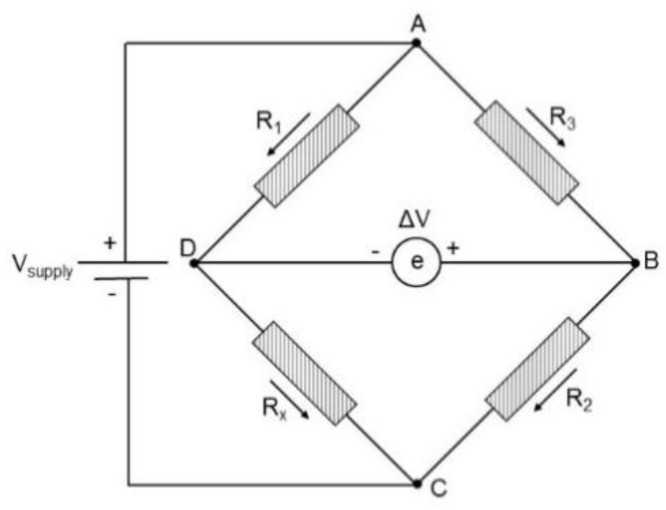
Wheatstone bridge.

**Figure 2 sensors-21-00172-f002:**
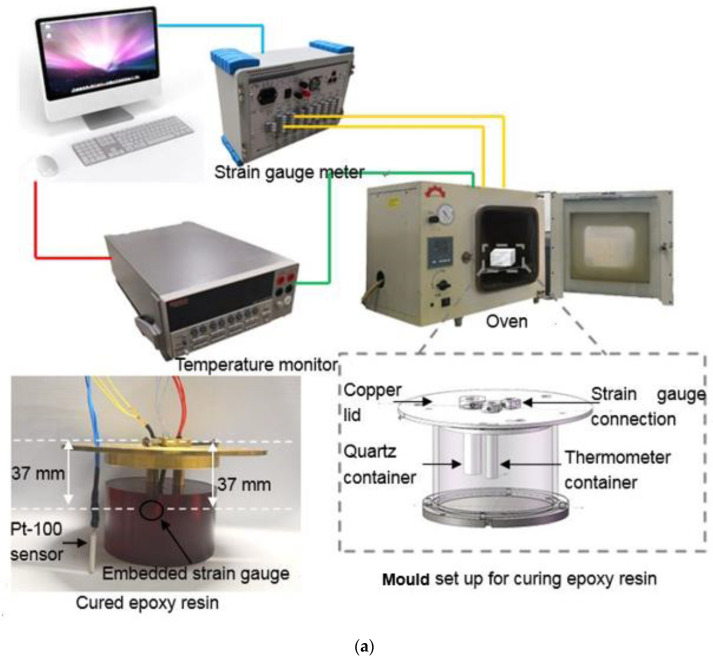
(**a**) schematic diagram of the test setup, (**b**) the stainless steel mould, and (**c**) the silicone mould.

**Figure 3 sensors-21-00172-f003:**
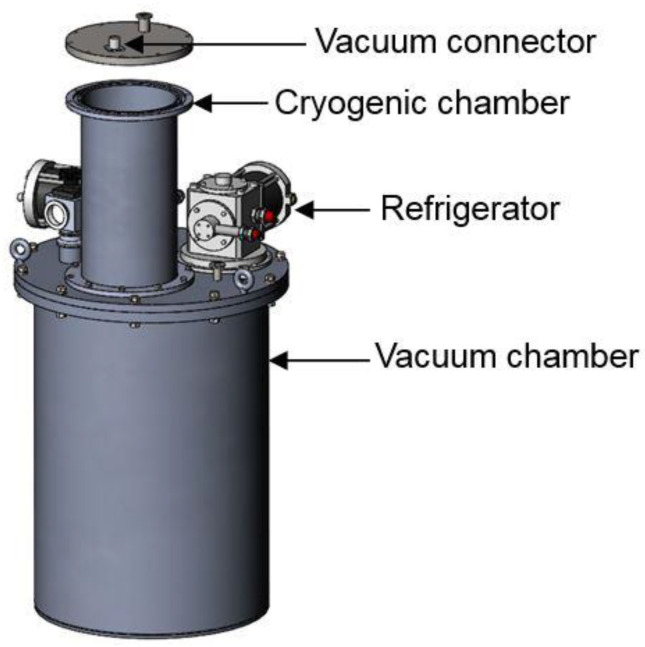
Cryogenic chamber to measure the thermal expansion coefficient of epoxy resin.

**Figure 4 sensors-21-00172-f004:**
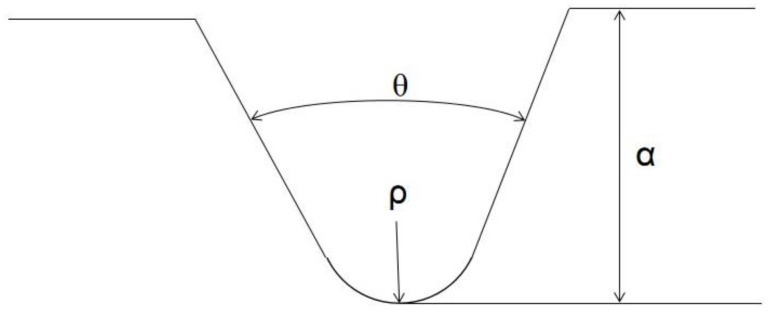
The notch draft of the specimen.

**Figure 5 sensors-21-00172-f005:**
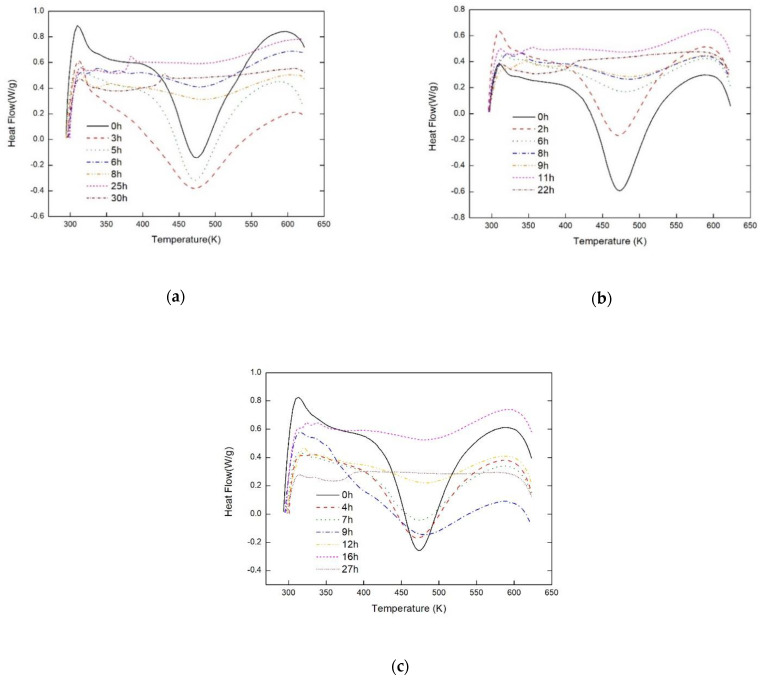
Non--isothermal DSC curves of curing reaction for epoxy system on different processes, (**a**) Process I, (**b**) Process II, and (**c**) Process III.

**Figure 6 sensors-21-00172-f006:**
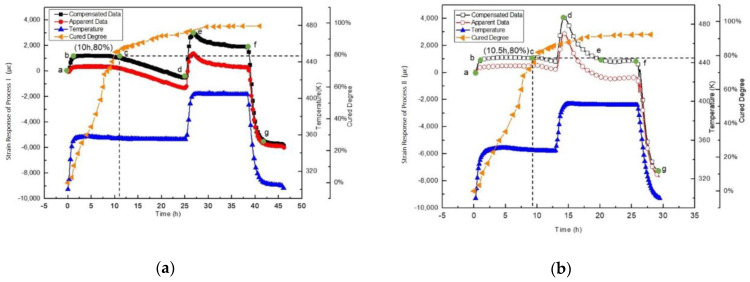
Curing degrees and comparison of real-time strain response of epoxy curing in different processes, apparent data and compensated data of (**a**) Process I, (**b**) Process II, and (**c**) Process III, and (**d**) strain responses (compensated data) comparison of three curing processes.

**Figure 7 sensors-21-00172-f007:**
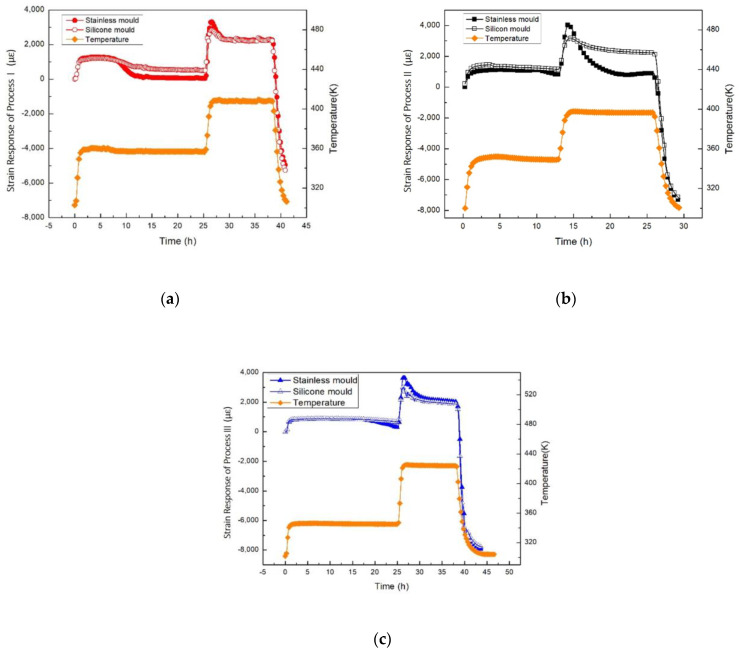
Strain responses of epoxy resin cured in two moulds: (**a**) Strain comparison of Process I in two moulds; (**b**) Strain comparison of Process II in two moulds; (**c**) Strain comparison of Process III in two moulds.

**Figure 8 sensors-21-00172-f008:**
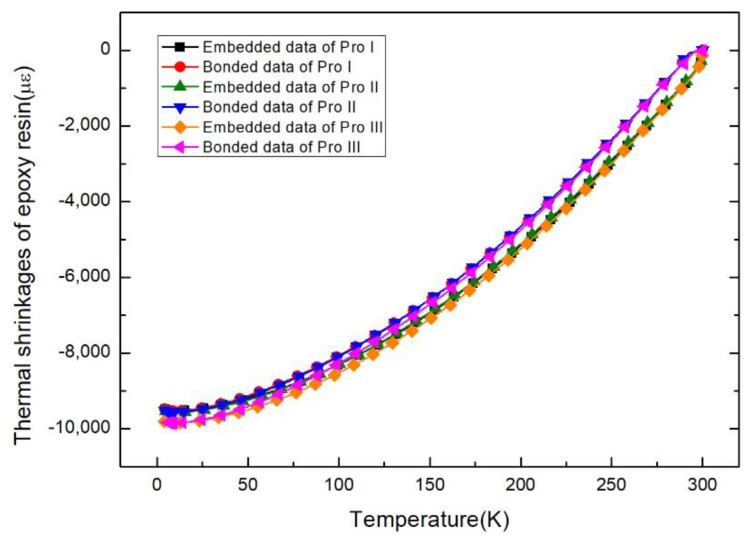
Thermal shrinkages of epoxy resin during cooling down to cryogenic temperature with three curing processes.

**Figure 9 sensors-21-00172-f009:**
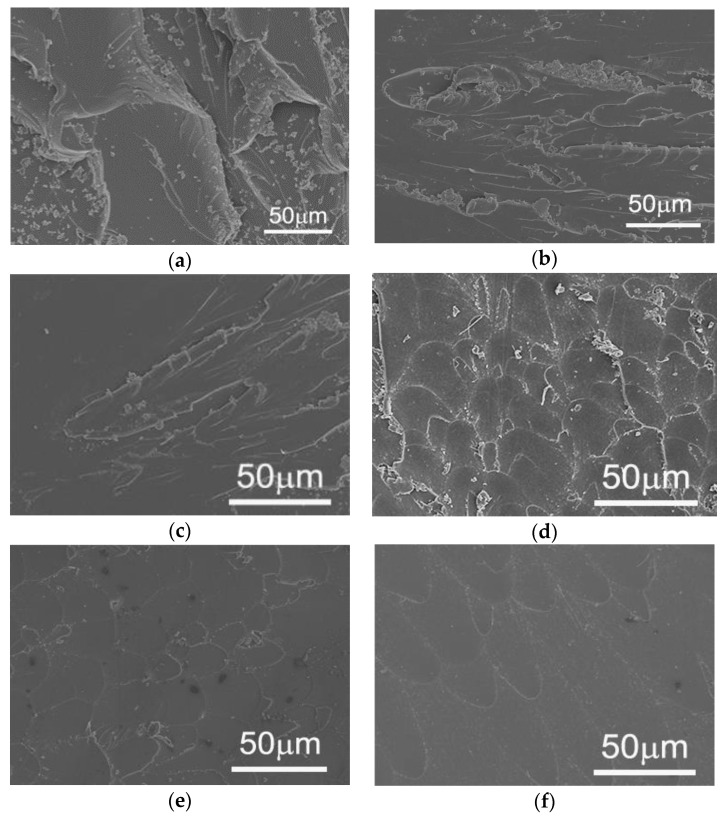
SEM images of epoxy resin fractured at room and cryogenic temperatures, (**a**) cured with Pro. I and fractured at room temperature, (**b**) cured with Pro. II and fractured at room temperature, (**c**) cured with Pro. III and fractured at room temperature, (**d**) cured with Pro. I and fractured at cryogenic temperature, (**e**) cured with Pro. II and fractured at cryogenic temperature, and (**f**) cured with Pro. III and fractured at cryogenic temperature.

**Table 1 sensors-21-00172-t001:** Cured percentage and Residual strains of epoxy resin curing with different processes.

Curing Process	Cured Percentage/%	Residual Strain/με
Pro. I: 353 K/24 h + 403 K/12 h	98.25	5945
Pro. II: 353 K/12 h + 403 K/12 h	92.44	17,169
Pro. III: 343 K/24 h + 423 K/12 h	87.65	8183

**Table 2 sensors-21-00172-t002:** Impact strengths of epoxy resins with three curing processes.

Curing Process	Impact Strength at RT kJ/m^2^	Impact Strength at 77 K kJ/m^2^
Pro. I	31.0 ± 2.3	15.4 ± 1.6
Pro. II	20.1 ± 1.6	11.8 ± 1.1
Pro. III	19.7 ± 2.4	11.5 ± 1.8

## Data Availability

The data presented in this study are available on request from the corresponding author.

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
