# Peer review of "Monitoring Strain Response of Epoxy Resin during Curing and Cooling Using an Embedded Strain Gauge"

_sensors, 2020, doi:10.3390/s21010172_

Round 1

Reviewer 1 Report

Exponent in power terms such as units and numbers needs to be corrected (e.g. line 74, 80, 83, etc). Applies to the entire manuscript;

Spaces added in line 79 also spaces missing before citations brackets (e.g. ISO 8130–6:1992[28]). Applies at many places of manuscript;

Please explain 'apparent data' and 'compensated data' as well as any the relationship between them in the manuscript text Figure 8,

please use same colors for bonded and embedded in all cases pro I, II and II Figure 9 needs to be enhanced (contrast).

Reviewer 2 Report

The work is well presented, but below are my comments:

  1. In the Introduction Chapter, wherever the literature is cited, the first bracket should be separated from the previous word. (line 31, 34, 35, 37, 39, 41,......) for example: parts[4] must be parts [4].

  1. When CTE is first mentioned it would be better to write its true sign: alpha.

  1. Line 79: metalic and polymeric - there are too many spaces between words.

  1. In lines 91 and 94 it is better to write only the first author and then write et al.

  1. In line 145, for the equation 2-2 authors should write what alpha means.

  1. Figure 2: correct the word mold in mould.

  1. Be sure to write the space between the size and the unit of measurement, eg line 184: 10mm, should be 10 mm.

  1. Chapter 3.1. needs to be rearranged. At the moment, it is quite difficult for the reader to follow the marks a, b, c....... while all this is two pages later in the picture. The description of the text must be just before that picture (picture number 6.d).

  1. Line 233 says they are not identical. When you look at Figures 7 the diagrams are the same.

  1. Do not separate Figure 8a and b from Figure 8c. Figure c should be merged to be all on the same page.

  1. The chapter Results must be separate from the chapter Discussion. In order to raise this paper scientifically, it is necessary to refer to the previous literature in the chapter Discussion to see what has improved with your method of measurement, etc....
